# Post-Neoadjuvant Surveillance and Surgery as Needed Compared with Post-Neoadjuvant Surgery on Principle in Multimodal Treatment for Esophageal Cancer: A Scoping Review

**DOI:** 10.3390/cancers13030429

**Published:** 2021-01-23

**Authors:** Julian Hipp, Blin Nagavci, Claudia Schmoor, Joerg Meerpohl, Jens Hoeppner, Christine Schmucker

**Affiliations:** 1Center of Surgery, Department of General and Visceral Surgery, Medical Center-University of Freiburg, Faculty of Medicine, University of Freiburg, 79106 Freiburg, Germany; julian.hipp@uniklinik-freiburg.de; 2Institute for Evidence in Medicine (for Cochrane Germany Foundation), Medical Center-University of Freiburg, Faculty of Medicine, University of Freiburg, 79110 Freiburg, Germany; nagavci@ifem.uni-freiburg.de (B.N.); meerpohl@ifem.uni-freiburg.de (J.M.); 3Clinical Trials Unit, Medical Center-University of Freiburg, Faculty of Medicine, University of Freiburg, 79110 Freiburg, Germany; claudia.schmoor@uniklinik-freiburg.de; 4Cochrane Germany, Cochrane Germany Foundation, 79110 Freiburg, Germany; 5Department of Surgery, University Medical Center Schleswig-Holstein, UKSH Campus Lübeck, 23538 Lübeck, Germany; jens.hoeppner@uksh.de

**Keywords:** esophageal cancer, watch-and-wait, surveillance and surgery as needed, esophagectomy, pathological complete response, neoadjuvant treatment

## Abstract

**Simple Summary:**

A substantial fraction of patients with esophageal cancer show post-neoadjuvant pathological complete response (pCR). Principal esophagectomy after neoadjuvant treatment is the standard of care for all patients, including those with pCR. Surveillance and surgery as needed may be a treatment alternative for these patients. We performed a scoping review and described all relevant clinical studies addressing these two treatment approaches. We identified three completed randomized controlled trials (RCTs) including 468 participants, three planned/ongoing RCTs with a planned sample size of 752 participants, one non-randomized controlled study with 53 participants, ten retrospective cohort studies (2228 participants) and one survey on patients’ preferences (100 participants). The current scoping review reveals that although surveillance and surgery as needed has been investigated within different study designs, the available study pool show methodological limitations and clinical results are heterogeneous. A thoroughly planned RCT considering these limitations will be of great importance to provide these patients with the best treatment.

**Abstract:**

Background: A substantial fraction of patients with esophageal cancer show post-neoadjuvant pathological complete response (pCR). Principal esophagectomy after neoadjuvant treatment is the standard of care for all patients, although surveillance and surgery as needed in case of local recurrence may be a treatment alternative for patients with complete response (CR). Methods: We performed a scoping review to describe key characteristics of relevant clinical studies including adults with non-metastatic esophageal cancer receiving multimodal treatment. Until September 2020, relevant studies were identified through systematic searches in the bibliographic databases Medline, Web of Science, Cochrane Library, Science Direct, ClinicalTrials, the German study register, and the WHO registry platform. Results: In total, three completed randomized controlled trials (RCTs, with 468 participants), three planned/ongoing RCTs (with a planned sample size of 752 participants), one non-randomized controlled study (NRS, with 53 participants), ten retrospective cohort studies (with 2228 participants), and one survey on patients’ preferences (with 100 participants) were identified. All studies applied neoadjuvant chemoradiation protocols. None of the studies examined neoadjuvant chemotherapeutic protocols. Studies investigated patient populations with esophageal squamous cell carcinoma, adenocarcinoma, and mixed cohorts. Important outcomes reported were overall, disease-free and local recurrence-free survival. Limitations of the currently available study pool include heterogeneous chemoradiation protocols, a lack of modern neoadjuvant treatment protocols in RCTs, short follow-up times, the use of heterogeneous diagnostic methods, and different definitions of clinical CR. Conclusion: Although post-neoadjuvant surveillance and surgery as needed compared with post-neoadjuvant surgery on principle has been investigated within different study designs, the currently available results are based on a wide variation of diagnostic tools to identify patients with pCR, short follow-up times, small sample sizes, and variations in therapeutic procedures. A thoroughly planned RCT considering the limitations in the currently available literature will be of great importance to provide patients with CR with the best and less harmful treatment.

## 1. Introduction

Currently in western Europe, the majority of patients with non-metastatic resectable esophageal cancer (EC) undergo neoadjuvant treatment followed by surgery on principle. Neoadjuvant treatment can either be neoadjuvant chemotherapy (nCT) or neoadjuvant chemoradiation (nCRT) for patients with esophageal adenocarcinoma (EAC) or nCRT for patients with esophageal squamous cell carcinoma (ESCC) [1]. Despite substantial progress in surgical technique and perioperative treatment, esophagectomy still implicates postoperative mortality rates between 4 and 11% and postoperative morbidity rates ranging between 36 and 80% [2,3,4,5]. Neoadjuvant treatment has become increasingly effective in recent years and post-neoadjuvant pathological complete response rates (pCR) between 16 and 49% are reported with current neoadjuvant treatment protocols. A higher pCR rate is observed in patients with ESCC with up to 49% compared to patients with EAC ranging between 16 and 35% [6,7,8,9]. Furthermore, several studies investigated the pCR rates for different current treatment protocols [6,8,9]. Retrospective studies have suggested the feasibility of omission of surgery in pCR patients without compromising survival rates [10,11]. The efficacy of a specific diagnostic protocol for clinical response revaluation has been only recently evaluated in patients with EC [12].

These observations impose the ethical need to identify patients undergoing potentially unnecessary esophagectomy. We, therefore, aim to establish a treatment/follow-up protocol to identify patients with post-neoadjuvant pCR and omit esophagectomy by close-meshed active surveillance and surgery as needed only in case of tumor recurrence. The current scoping review is part of the development phase for a planned prospective multicenter randomized controlled trial addressing the issue of “surgery as needed versus surgery on principle in patients with post-neoadjuvant complete response of esophageal cancer” (preliminary prospective registration identifier of the clinical trial: DRKS 00022801). The scoping review will address the following questions: What specific neoadjuvant protocols of nCRT and nCT have been studied for surveillance and surgery as needed?In which populations or settings have these protocols been studied?Which diagnostic methods have been used for post-neoadjuvant tumor staging and surveillance of tumor response?Which outcomes have been addressed in the published studies on surveillance and surgery as needed in EC patients?Which trial designs have been used?Which results were observed with respect to the disease-free survival (DFS) and overall survival (OS) rates in already completed randomized controlled trials (RCTs)?Which problems occurred with respect to recruitment and compliance in already completed RCTs?

The results of the scoping review will allow us to finally define and adapt the research question including the design and methodology of the RCT.

## 2. Results

We identified 703 titles and abstracts; for 81 of these, the full text was evaluated. The Preferred Reporting Items for Systematic Reviews and Meta-Analyses (PRISMA) flow diagram (Figure 1) outlines the screening and selection process of these articles. (Table 1, Table 2 and Table 3) presents the key characteristics of the identified six RCTs, one non-randomized controlled study (NRS), ten observational studies and one survey on patients’ preferences investigating surveillance after neoadjuvant treatment of EC.

### 2.1. Randomized Controlled Trials (RCTs)

#### 2.1.1. Setting, Key Characteristics, and Patient Population

The key characteristics of the three RCTs (ESOPRESSO [A Randomized Phase III Trial on the Role of Esophagectomy in Complete Responders to Preoperative Chemoradiotherapy for Esophageal Squamous Cell Carcinoma], FFCD (Fédération Francophone de Cancérologie Digestive) 9102 trial, and Stahl et. al. [16]) are displayed in Table 1. Of the 468 patients randomized, 234 patients received surgery on principle (control group) and 234 patients were randomized to the surgery as needed group (intervention group) [13,14,16]. Two trials were conducted in Germany and France in the 1990s and published in 2005 and 2006 respectively, and another recently published trial (in 2019) was conducted in South Korea. The European trials were multicenter trials with 259 patients [14] and 172 patients [16] respectively, while the South Korean trial was a single center RCT with only 37 randomized patients [13]. Median follow-up was 29.9 [13], 24 [14], and 60 months [16].

In the Korean ESOPRESSO trial [13] (thoracic ESCC) and in the German trial conducted by Stahl et al. [16] (tumor in the upper or mid third of the esophagus), only patients with locally advanced ESCC were included. The French FFCD 9102 trial [14] included both entities (ESCC and EAC), although ESCC patients predominated in this trial with 88.8% of the study population. The mean patient age ranged between 57 and 62 years. 

The three RCTs differ in regard to the design aspects. In the ESOPRESSO trial, superiority of surgery vs. surveillance with respect to DFS is hypothesized. The FFCD 9102 trial and Stahl et al. [16] aim to show equivalence of surveillance vs. surgery on principle, but sample size calculations are based on different equivalence margins (FFCD 9102: 10% in 2-year OS rate; Stahl et al. [16]: 15% in 2-year OS rate) (Table 4).

#### 2.1.2. Neoadjuvant Treatment Protocols

All three trials investigated neoadjuvant chemoradiation protocols. No RCT was performed with a neoadjuvant chemotherapy protocol. In the ESOPRESSO trial, patients received induction chemotherapy with Capecitabine and Cisplatin and proceeded to chemoradiation with Capecitabine and Cisplatin and 28 × 1.8 Gy (total dose: 50.4 Gy) in both arms. In the FFCD 9102 trial, patients received two sequences of split-course radiotherapy with daily fractions of 3 Gy (total dose: 30 Gy) or conventional radiotherapy with daily fractions of 2 Gy for 4.5 weeks (total dose: 46 Gy). Two cycles of chemotherapy (Cisplatin and 5-Flourouracil (5-FU)) were administered during neoadjuvant radiotherapy. In the surveillance-arm of the FFCD 9102 trial, chemoradiation was continued with one sequence of split-course radiotherapy (total dose of complete radiotherapy: 45 Gy) or continuation of conventional radiotherapy with additional 20 Gy (total dose of complete radiotherapy: 66 Gy). Each radiotherapeutic option was combined with three cycles of chemotherapy with Cisplatin and 5-FU. In the trial conducted by Stahl et al. [16], patients underwent a chemoradiation protocol with three cycles of FLEP-chemotherapy (5-FU, Leucovorin, Etoposide, and Cisplatin) followed by chemoradiation with Cisplatin and Etoposide and 40 Gy radiotherapy in 4 weeks with daily doses of 2 Gy prior to surgery. In the intervention-arm without surgery, treatment consisted of the same induction chemotherapy and combined chemoradiation, but afterwards, the radiation dose was increased to at least 65 Gy.

#### 2.1.3. Surveillance vs. Surgical Treatment

In all RCTs, patients were included/recruited before neoadjuvant treatment started. Randomization was carried out after the response evaluation in the ESOPRESSO and FFCD 9102 trials in case of complete clinical response (cCR), and clinical partial response in the FFCD 9102 trial, either to immediate surgery or surveillance with surgery as needed. Stahl et al. [16] randomized all patients before treatment either to neoadjuvant chemoradiation and mandatory surgery or to definitive chemoradiation. Salvage surgery was performed in case of unresponsive tumors after definitive chemoradiation. The interval between chemoradiation and surgery ranged from 2–5 weeks [16] to 7–9 weeks [14].

#### 2.1.4. Diagnostic Methods for Post-Neoadjuvant Tumor Staging and Surveillance of Tumor Response

In the ESOPRESSO trial, pretreatment staging was performed with esophagogastroduodenoscopy with biopsy, thoracic/abdominal/pelvic computed tomography (CT), endoscopic ultrasonography, bone scan, 18F-fluorodeoxyglucose (FDG)-positron emission tomography (PET), and bronchoscopy when needed. Four weeks after completing chemoradiation, patients were reevaluated by endoscopy with biopsy, chest CT, and positron emission tomography-computed tomography (PET-CT). cCR was defined as no radiographic or metabolic evidence of disease without residual tumor on endoscopy with biopsy. Complete metabolic response was defined as complete resolution of FDG-uptake within all lesions. Otherwise, the response was assessed according to Response Evaluation Criteria in Solid Tumors (RECIST) criteria [29]. Only patients who achieved cCR after chemoradiation were randomized. In the FFCD 9102 trial, the initial staging included clinical examination, esophagogastroduodenoscopy with biopsies, esophagogram, bronchoscopy, supraclavicular ultrasonography, thoracoabdominal CT scan, and endoscopic ultrasonography when available. After neoadjuvant treatment, cCR was defined by the absence of dysphagia and of visible tumor on the post-neoadjuvant esophagogram. A partial response was defined as a decrease of more than 30% of the tumor length on the post-neoadjuvant esophagogram and improvement of dysphagia. In the absence of objective response or in case of contraindication to surgery, the treatment was decided by the investigator. All patients with cCR or partial response and good toleration of chemoradiation were randomly assigned to surgery or continuation of chemoradiation. In the trial conducted by Stahl et al. [16], criteria for cCR were defined as no dysphagia, normal barium esophagogram and esophagogastroduodenoscopy, and normal CT scan. Partial remission was defined as improvement of dysphagia, greater than 50% tumor regression evaluated by CT and greater than 50% reduction of intraesophageal tumor extension as assessed by barium swallow.

#### 2.1.5. Follow-Up Visits

Follow-Up visits were performed in all trials every 3 months for 2 years and then every 6 months up to 5 years after treatment. In the ESOPRESSO trial, patients were followed-up by CT scan and endoscopy every 6 months. In the FFCD 9102 trial, the follow-up was performed by endoscopy with biopsies, esophagogram, thoracoabdominal CT scan and, if available, endoscopic ultrasonography. Dysphagia was scored according to the O’Rourke criteria [30]. Patterns of first recurrence (locoregional, distant, or both), hospitalizations, and palliative procedures against dysphagia were reported. Stahl et al. [16] do not describe details on follow-up-assessments. 

#### 2.1.6. Outcomes

All three RCTs reported OS as an outcome. It was the primary outcome in the FFCD 9102 and in the trial of Stahl et al. [16], while the ESOPRESSO trial used DFS as the primary outcome. Besides OS, the reported outcomes are heterogeneous and shown in Table 4. No relevant difference was observed regarding OS in all three trials between the treatment groups, while the local progression-free survival (PFS) was favorable in the surgical group in the Stahl and FFCD 9102 trial and a trend towards a better DFS was observed in surgical patients in the ESOPRESSO trial. pCR rates ranged between 35% and 69%. Surgery during surveillance was performed in 6% of patients in the Stahl trial and in 33% in the ESOPRESSO trial.

Remarkable differences in the adherence to the assigned treatment could be observed in all trials. While compliance was good in the surveillance-arms, a high-rate of non-compliance was observed in the surgical-arms of all RCTs and led to early study closure of the ESOPRESSO trial (Table 4). 

### 2.2. Non-Randomized Controlled Studies (NRS)

#### 2.2.1. Setting, Key Characteristics, and Patient Population

The characteristics of the single NRS are shown in Table 1. The study was conducted between 1994 and 2002 in Japan. A total of 124 patients were screened and 53 patients were finally included [17]. Median follow-up time was 51 months. Patients with cT4, cN0-1, and cM0 ESCC of the thoracic esophagus were included. No information was available on the age of the patients.

#### 2.2.2. Neoadjuvant Treatment Protocols

The neoadjuvant treatment included chemoradiation with cisplatin and 5-FU combined with 36 Gy radiotherapy (1.2 Gy per day). After this first cycle, patients received surgery or were treated after a surveillance protocol. Both groups of patients underwent the same cycle of chemoradiation for a second time, either immediately or one month after surgery.

#### 2.2.3. Surveillance vs. Surgical Treatment

This study was a NRS based on the informed decision that patients chose whether to undergo surgery (control group) between the first and second chemoradiotherapy cycle or surveillance (intervention group). Twenty-three patients opted for surveillance and surgery as needed, while 30 patients did undergo immediate surgery. Salvage surgery was performed as needed (in one case). 

#### 2.2.4. Diagnostic Methods for Post-Neoadjuvant Tumor Staging and Surveillance of Tumor Response

The pretreatment staging evaluation consisted of a general physical examination, chest and abdominal radiography, esophagogram, esophagogastroduodenoscopy, cervical and upper abdominal ultrasonography, CT scan of the neck, chest, and upper abdomen, magnetic resonance of imaging of the neck and chest, and a bone scintigraphy. Bronchoscopy was performed only for a cancer in the upper or middle thoracic esophagus. Evaluation of clinical response was performed using an esophagogram, esophagogastroduodenoscopy, and a CT scan. The response was considered complete when no radiographic evidence of disease was seen, no residual tumor was found during esophagogastroduodenoscopy, and the biopsy was negative.

#### 2.2.5. Follow-Up Visits

Follow-up using a general physical examination, tumor markers, and chest radiographs were performed every month for the first 2 years, every 2 months for 2 to 3 years after treatment, every 3 months for 3 to 5 years after treatment, and every 6 months thereafter. Endoscopy, ultrasonography of the neck and abdomen, CT scan, and bone scintigraphy were routinely scheduled every year and repeated when any new clinical symptoms appeared or if any of the tumor markers increased to an abnormal level.

#### 2.2.6. Outcomes

Reported outcomes in this study are OS for the entire cohort, for responders and non-responders, response rates of chemoradiation, therapeutic toxicity, and postoperative complications. OS did not differ between responders to chemoradiation with and without esophagectomy, while non-responders did benefit from radical surgery. The pCR rate after the first cycle of nCRT was only 7%, although 60% of the patients were classified as clinical responders. 

### 2.3. Planned/ongoing Randomized Controlled Trials (RCTs)

#### 2.3.1. Setting, Key Characteristics, and Patient Population

Three planned/ongoing RCTs were identified (Table 1). SANO (Surgery As Needed approach in Oesophageal cancer patients) is a phase III multicenter non-inferiority trial with a stepped-wedge clustered design conducted in the Netherlands with the planned recruitment of 600 patients [18]. ESOSTRATE (Comparison of Systematic Surgery Versus Surveillance and Rescue Surgery in Operable Oesophageal Cancer With a Complete Clinical Response to Radiochemotherapy) is a phase III multicenter trial conducted in France with a planned recruitment of 600 patients (NCT02551458) and CELAEC is a Chinese phase III multicenter trial with a planned recruitment of 196 patients [19]. 

The SANO and ESOSTRATE trials consider patients with resectable, locally-advanced ESCC and EAC of the esophagus and the esophagogastric junction (AEG I-II-tumors). In the CELAEC trial, Chinese patients with resectable ESCC are eligible. 

The design of these trials differs: The SANO trial has a cluster-randomization and the ESOSTRATE trial a randomization of patients with cCR, while patients will be randomized before the start of the nCRT in the CELAEC trial. Furthermore, these trials differ in the definition of the primary endpoints, trial hypotheses, and assumptions for sample size calculations. The SANO trial aims to demonstrate non-inferiority of surveillance vs. surgery on principle (non-inferiority margin 15% in 3-year OS rate), the CELAEC trial aims to show superiority of surveillance vs. surgery on principle with respect to OS (5-year OS rate: 29.4% surgery vs. 50% surveillance) and the ESOSTRATE trial hypothesizes superiority of surgery on principle vs. surveillance with respect to DFS (2-year DFS rate: 45% surgery vs. 30% surveillance) (Table 4).

#### 2.3.2. Neoadjuvant Treatment Protocols

Patients will be treated according to the CROSS protocol (Carboplatin, Paclitaxel, 41.4 Gy in 23 fractions) in the SANO trial. The ESOSTRATE trial includes patients with cCR after nCRT without specifying a treatment protocol. The CELAEC trial randomizes patients either to definitive chemoradiation (dCRT) or neoadjuvant chemoradiation (nCRT) followed by surgery on principle. The dCRT group is treated with 50 Gy in 25 fractions and the nCRT group is treated with 42 Gy in 21 fractions. Patients will additionally be randomized to one of three chemotherapy protocols during chemoradiation (XELOX, Capecitabine mono or Cisplatin/5-FU).

#### 2.3.3. Surveillance vs. Surgical Treatment

Patients with cCR during the second clinical response evaluation will be randomly assigned to active surveillance (intervention arm) or standard surgery (control arm) in the SANO trial. This trial has a special study design: as the authors expect difficulties with the patient’s cooperation towards randomized treatment modality, the stepped-wedge cluster randomized design involves the random sequential switch of clusters of participating institutions from the surgical-arm towards the active surveillance-arm. Therefore, the treatment decision will be made for the patient due to the status of the trial center in which the patients want to be treated. Patients therefore will not need to accept the result of randomization for their individual treatment. Salvage surgery will be offered to the patients in case of local recurrence and palliative care will be offered in case of disseminated disease during surveillance. Patients in the ESOSTRATE trial will be randomly assigned either to surgery on principle or surveillance and surgery as needed in cases of resectable loco-regional recurrence. In the CELAEC trial, patients are randomly assigned to definitive chemoradiation or neoadjuvant chemoradiation with surgery. Patients with resectable disease in the dCRT group can be treated with esophagectomy, if cCR is not reached within 16 weeks after treatment or in case of resectable local recurrence.

#### 2.3.4. Diagnostic Methods for Post-Neoadjuvant Tumor Staging and Surveillance of Tumor Response/Follow-Up Visits

In the SANO trial, patients will undergo a first clinical response evaluation including esophagogastroduodenoscopy with at least 8 (random) biopsies, including at least 4 bite-on-bite biopsies of the primary tumor site and of any other suspected lesions 4–6 weeks after completion of nCRT. Patients with evidence of locoregional residual disease during first clinical response evaluation will be offered a subsequent 18F–FDG PET-CT to exclude disseminated disease and will be offered immediate surgery. Patients who are found to be cCR will undergo a second clinical response evaluation 6–8 weeks after first clinical response evaluation. This second clinical evaluation will include an 18F–FDG PET-CT, followed by esophagogastroduodenoscopy with bite-on-bite biopsies of the primary tumor site and any other suspected lesions, radial endoscopic ultrasound and in case of PET-positive lesions and/or suspected lymph nodes linear endoscopic ultrasound with fine needle aspiration (FNA) cytology. Patients with evidence of locoregional residual disease or highly suspected locoregional residual disease on 18F–FDG PET-CT and without distant metastases during clinical response evaluation will undergo immediate surgery. Patients with distant metastases will be referred for palliative care. Patients with cCR during second clinical response evaluation will be assigned to active surveillance (experimental arm) or standard surgery (control arm), according to the randomization. Patients in the active surveillance arm will undergo 18F- PET-CT, esophagogastroduodenoscopy with biopsies, including at least 4 bite-on-bite biopsies and endoscopic ultrasound with FNA at 6, 9, 12, 16, 20, 24, 30, 36, 48, and 60 months after neoadjuvant treatment [18]. 

Details on the specific clinical response evaluation protocol during the ESOSTRATE trial are not publicly available. Evaluation of the response will take place 5–6 weeks after completion of nCRT and patients with cCR will be randomized to surveillance and salvage surgery in cases of resectable loco-regional recurrence or surgery on principle. In the CELAEC trial, patients will be examined with esophagogastroduodenoscopy, endoscopic ultrasonography, CT scan of the thorax and abdomen with contrast and ultrasonography of the cervical region with FNA cytology for any suspicious nodes. PET-CT will only be used in case of suspected metastases. Details on follow-up examinations in this trial are not specified in the trial protocol, recurrence of the disease is defined as either endoscopic recurrence confirmed with biopsy or distant metastasis [19].

#### 2.3.5. Outcomes

Reported outcomes of these trials are shown in Table 4. The primary outcome in the SANO and CELAEC trial is OS and DFS in the ESOSTRATE trial. Besides survival-outcomes, important outcomes of the SANO trial are the percentage of patients in active surveillance without surgery, Health-related quality of life in both arms and the rate of irresectable recurrence during active surveillance. 

### 2.4. Observational Studies

#### 2.4.1. Setting, Key Characteristics, and Patient Population

The study characteristics of the ten observational studies are shown in Table 2. The identified observational studies were published between 1996 and 2020. Studies were performed in Europe (*n* = 4), North America (*n* = 5), and Australia/New Zealand (*n* = 1). Median follow-up ranged from 25.6–60 months.

Patient populations of these studies are heterogeneous. Two studies [10,21] only included patients with ESCC, one study only included patients with EAC [24], one study included only patients with the rare adenosquamous histological subtype of EC [20], and six studies included patients with EAC and ESCC [11,22,23,25,26,27]. Median patient ages ranged from 58.8–75 years. 

#### 2.4.2. Neoadjuvant Treatment Protocols

Treatment protocols included different chemoradiation protocols with a total dose of 30–60 Gy radiation. Protocols included either neoadjuvant [10,11,22,23,26] or definitive chemoradiation [21]. Two register studies did not specify chemoradiation [20,25] and one study included different protocols for neoadjuvant and definitive/palliative treatment [27]. No study included chemotherapeutic protocols without radiation.

#### 2.4.3. Surveillance vs. Surgical Treatment

Intervention arm (surveillance and surgery as needed) and control arm (surgery on principle) were arranged differently between the studies. Castoro et al., Taketa et al., van der Wilk et al., Furlong et al., and Murphy et al. informed the patients on their CR status after re-staging examinations. Surveillance was performed either due to patients’ choice or when a patient was considered unfit for surgery [10,11,22,23,24]. Wilson et al. stratified patients according to the post-neoadjuvant staging examination. In case of cCR, patients underwent surveillance and surgery on demand, while patients with residual tumor after neoadjuvant treatment were operated immediately. Münch et al. and Denham et al. compared dCRT and nCRT in ESCC without a therapeutic consequence being triggered by the CR status [21,27]. The reasons for surgical treatment or surveillance remained unclear in the cancer registry analyses performed by McKenzie et al. and Gamboa et al. [20,25].

#### 2.4.4. Diagnostic Methods for Post-Neoadjuvant Tumor Staging and Surveillance of Tumor Response

Response evaluation and definition of response and cCR is different between studies and hardly comparable. Van der Wilk et al. used the clinical response evaluation during the preSANO trial for their retrospective analysis with close meshed surveillance as stated in the protocol of the SANO trial [22]. Furlong et al., Taketa et al., and Castoro et al. used a similar approach with negative biopsies of the tumor bed and a negative computerized tomography (CT) or PET-CT to diagnose cCR [10,11,23]. In the study of Wilson et al., patients with negative biopsy and >75% regression on CT scan were classified as cCR and remained under endoscopic surveillance at intervals of 3 and 6 months in the first and second years of follow-up [26].

Murphy et al. defined CR as no evidence of local or distant disease progression on post-neoadjuvant PET-CT, negative post-neoadjuvant biopsy, and decrease in local standard uptake value (SUV) on PET-CT ≥35% after neoadjuvant treatment [24]. Assessment of cCR rate before surgical treatment was not performed in most studies [21,25,27].

#### 2.4.5. Outcomes

Outcomes of the identified studies are shown in Table 5. Almost all studies reported OS rates as an outcome. Several studies reported non-inferiority of surveillance compared to surgical resection after nCRT. Van der Wilk et al. [22] demonstrated in a propensity score matching of preSANO patients that patients undergoing active surveillance do not have a worse 3-year OS with 77% compared to patients with nCRT and surgery with 55%. The 3-year PFS was non-inferior with a trend towards a worse PFS in the surveillance group with 60 vs. 54%. Distant dissemination rate was equal in both groups (28%) [22]. Similar results are reported by Castoro et al. [10] (5-year-OS-surveillance: 57% vs. nCRT + surgery: 50%; 5-year-DFS-surveillance: 35% vs. nCRT + surgery: 56%), Furlong et al. [23] (median OS-surveillance: 55 months vs. nCRT + surgery: 56 months), and Taketa et al. [11] (median OS-surveillance: 58 months vs. nCRT + surgery: 51 months; recurrence-free survival-surveillance: 19 months vs. nCRT + surgery: 27 months). A (non-significant) trend towards improved OS in patients with surgery on principle was observed in the study by Münch et al. comparing nCRT with surgery and dCRT in ESCC patients (median OS-dRCT: 43 months vs. nCRT + surgery: 21 months) [21]. 

An improvement of OS with additional surgical resection after nCRT and cCR is reported by Murphy et al. [24] (median OS-surveillance: 21 months vs. nCRT+surgery: 46 months), although a relevant proportion of long-term survivors without surgery is observed and the authors conclude that “some patients may be destined for complete pathological response and are ultimately cured without surgery”. McKenzie et al. [25] concluded that surgical resection improves survival for ESCC patients (median OS-surveillance: 13 months vs. nCRT+surgery: 25 months) as well as for EAC patients (median OS-surveillance: 11 months vs. nCRT+surgery: 26 months). These results have to be interpreted carefully, as this study is prone to several biases. Neither a reason for omission of surgical treatment nor a grading of clinical response was reported. Further relevant outcomes were reported in other studies. pCR rates in the surgical groups following nCRT ranged between 20% and 69% [10,20,22]. Surgery during surveillance was performed in 16–48% of the patients [22,23]. These results have to be interpreted carefully, as most of these retrospective studies did not offer a real “active surveillance” with the option for salvage surgery as needed to all their patients as some were considered unfit for surgery or denied surgery. Therefore, a surgery rate of 48% as observed by van der Wilk et al. [22] seems to be the most appropriate approximation of the real-life anticipated requirement for surveillance and surgery on demand in a prospective randomized trial. 

### 2.5. Survey on Patients’ Preferences 

Noordman et al. [28] investigated factors influencing patients’ preferences and trade-offs that patients are willing to make when choosing between surgery and active surveillance after nCRT and cCR. Characteristics of this study are shown in Table 3 and Table 5. A discrete-choice experiment is a socioeconomic method which is used for the evaluation of the relative importance of different characteristics of a product/service and the rate at which individuals are willing to “trade” between [31]. By means of a discrete-choice experiment, the relative importance of five aspects (5-year OS, long and short-term quality of life, the risk of esophagectomy during surveillance, and the frequency of follow-up examinations) were evaluated. Five-year OS, the risk of esophagectomy during surveillance and long-term quality of life were the most important aspects. Astonishingly, patients were willing to trade-off substantial OS probability (16%) to avoid esophagectomy.

## 3. Discussion

We were able to identify six completed and planned/ongoing RCTs, one NRS, and ten observational studies exploring the possibilities of post-neoadjuvant surveillance and surgery as needed in clinical responders compared to the standard treatment with post-neoadjuvant surgery on principle in this scoping review.

With regard to the applied neoadjuvant treatment, the identified studies exclusively used nCRT protocols. Studies using nCT protocols could not be identified, although nCT according to the ECF or FLOT protocols are standard therapies in the western world for EAC patients, which also represent the majority of EC patients in western Europe [32]. Besides the complete absence of nCT protocols, there was a substantial heterogeneity of used nCRT protocols among the identified studies. Many studies used regimens with 5-FU and cisplatin combined with different doses of radiation therapy while the widely applied CROSS protocol was only used in two retrospective studies and in the ongoing SANO trial. Of note—as prospective data on this subject are missing—retrospective data suggest equivalence of these both regimens with a higher rate of pCR after treatment with 5-FU and cisplatin compared to the CROSS protocol [33,34,35]. The direct comparison of nCT (according to the FLOT protocol) and nCRT (according to the CROSS protocol) is also missing until the results of the ongoing ESOPEC trial (NCT02509286) will be available [36]. While the retrospective studies included (to some extent) current nCRT protocols, the two large European RCTs conducted in the 1990s used nCRT protocols that are nowadays outdated, which limits the transferability and applicability of these trial.

Regarding the study populations, the retrospective studies mainly investigated mixed populations with EAC and ESCC. The RCTs, however, did almost exclusively include ESCC patients. Consequently, the comparability to a mixed prospective cohort of EC patients in Europe is limited. 

The methods used for post-neoadjuvant tumor staging and surveillance are very heterogeneous across studies. While older studies were based on simple response evaluation with clinical manifestations like the grade of dysphagia and basic radiologic examinations as esophagograms were used [14], the assessment of cCR has become increasingly complex in recentstudies [12,13]. Although every study used its own protocol for post-neoadjuvant tumor staging and several different definitions of cCR were used across studies, the efficacy of diagnostic methods to detect cCR were only evaluated within one protocol. The most sophisticated and prospectively evaluated approach for restaging after neoadjuvant treatment and detection of residual disease and recurrence during surveillance is presented in the preSANO trial. The study carried out endoscopies with bite-on-bite biopsies and FNA of suspicious lymph nodes for the detection of locoregional residual disease, and PET-CT scan for detection of interval metastases. This method has been proven to show high diagnostic accuracy in the prospective c cohort study preSANO [12]. 

Overall, the reported primary outcomes are overall and disease-/recurrence-free survival. In the RCTs and most of the retrospective studies, OS was not compromised for patients undergoing surveillance but many of the studies reported decreased disease-free or recurrence-free survival. As the local recurrence during surveillance is anticipated to be a frequently occurring event, a reduced DFS is expected in these studies. More important than DFS time and rate is the probability of a local-recurrence being identified early enough to be resectable by surgery as needed without simultaneous distant disease dissemination. The respectability of recurrent cancer and an equal distant dissemination rate in patients during surveillance compared to patients after radical resection of EC were observed in several of the identified studies [10,11,22,23]. 

In total, we were able to identify six RCTs (completed and ongoing) comparing post-neoadjuvant surveillance and surgery as needed with post-neoadjuvant surgery on principle. Interestingly, all these trials, which essentially try to answer the same research question, have varying study designs with non-inferiority, superiority, and equivalence designs. Furthermore, different non-inferiority and equivalence margins and different primary endpoints with OS and DFS were used. From our point of view, a non-inferiority design is the appropriate design for an RCT to compare surveillance and surgery as needed with surgery on principle after neoadjuvant treatment of EC. In the published trials, a striking difference in the compliance to the allocated treatment was noticed between different trial arms, with higher rates of non-compliance to the protocol in the surgical-arms. Noordman et al. included this factor to their study design by using a cluster randomization. For our planned RCT, we are going to address this issue by conducting a pilot study to create patient-centered trial information material. This will be based on patients’ preferences regarding treatment options, which will be evaluated by a newly developed patient decision aid questionnaire. This will improve study recruitment and protocol adherence by creating comprehensive and patient-centered study information material.

One very interesting outcome of the survey on patients’ preferences was that patients differ in their preferences from physicians, who are used to aim for a maximization of survival rates. In contrast to this approach, patients are willing to trade off OS probability to a certain degree towards increased quality of life and omission of surgery [28].

It is important to emphasize the lack of high-quality data from RCTs with modern diagnostic methods for detection of cCR and with modern neoadjuvant treatment strategies (nCRT and especially nCT). Moreover, high-level evidence from RCTs is almost entirely missing for patients with EAC; therefore, there is an urgent need to perform further RCTs regarding surveillance and surgery as needed with modern diagnostic and multimodal treatment strategies. 

## 4. Materials and Methods 

### 4.1. Search Methods

This scoping review is based on a prospectively published protocol in “BMJ Open” (doi:10.1136/bmjopen-2020-) and is reported according to the PRISMA extension for scoping reviews (PRISMA-ScR) statement [37]. The searches for this scoping review were performed on 25th of September 2020. Search strategies do adhere to the recommendation of PRESS (Peer Review of Electronic Search Strategies) [38]. An expert medical sciences librarian developed the search strategies, which were then peer-reviewed (search strategies presented in Appendix A). Search strategies were validated by checking whether they identified studies already known, including RCTs identified by a Cochrane Review from 2016 [39]. We did not use any date restriction in the electronic searches. For each database, the date of the search, the search strategy, and the number of search results were documented. 

Systematic searches for relevant published trials were conducted in the following electronic data sources (Appendix A): Medline, Medline Daily Update, Medline In Process and Other Non-Indexed Citations, Medline Epub Ahead of Print (via Ovid)Web of Science Core Collection: Science Citation Index-EXPANDED (SCI-EXPANDED) (via Clarivate Analytics)Cochrane Library (via Wiley)Science Direct (via Elsevier).

Searches for ongoing or unpublished completed studies were performed in ClinicalTrials.gov (www.clinicaltrials.gov), the German study register (www.drks.de), and in the World Health Organization (WHO, Geneva, Switzerland) International Clinical Trials Registry Platform (ICTRP) (http://www.who.int/ictrp/search/en). 

We used relevant studies and/or systematic reviews to search for additional references via the PubMed similar articles function (https://www.nlm.nih.gov/bsd/disted/pubmedtutorial/020_190.html) and forward citation tracking. Reference lists of relevant studies and systematic reviews were also reviewed manually. Titles and abstracts of the records identified by the searches were screened by one reviewer and full texts of all potentially relevant articles were obtained. Full texts were checked for eligibility by two reviewers and reasons for exclusions were documented. The complete screening process was conducted in Covidence (https://www.covidence.org/).

### 4.2. Eligibility Criteria 

Population: Adults with any type of non-metastatic esophageal cancer (after receiving neoadjuvant treatment) were included. Distant metastasis due to EC, presence of gastric cancer, and patients under 18 years of age were not considered. 

Intervention and comparator treatment: This scoping review considered surveillance after neoadjuvant therapy as eligible intervention. We considered all neoadjuvant chemotherapeutic and neoadjuvant chemoradiotherapeutic interventions implemented and evaluated in the context of non-metastatic EC. Surgery on principle after neoadjuvant therapy was considered as the comparator treatment.

Outcomes: We captured the following outcomes: OS, PFS, proportion of radical resection margin (R0-resection), postoperative complications, rate and timing of distant dissemination, disease recurrence rate, pCR rate in surgical group, and rate of surgical treatment during surveillance. 

We included RCTs, NRS using strategies of non-random allocation for assigning interventions and observational studies (with control group) for the scoping review. We did not consider case reports, case series, review articles, clinical guidelines, and work that had not been peer-reviewed (e.g., thesis, editorials, letters, comments). We did not apply any exclusion criteria regarding study duration and/or the study setting. Risk of bias assessment is not part of a scoping review and was not assessed [40].

## 5. Conclusions

In conclusion, the available trials and studies suggest that post-neoadjuvant surveillance and surgery as needed is feasible for complete clinical responders without compromising OS. The retrospective data especially from van der Wilk, Taketa, Castoro, and Furlong et al. and the results of the preSANO trial justify a prospective randomized controlled phase III trial based on neoadjuvant treatment with modern nCT and nCRT protocols.

## Figures and Tables

**Figure 1 cancers-13-00429-f001:**
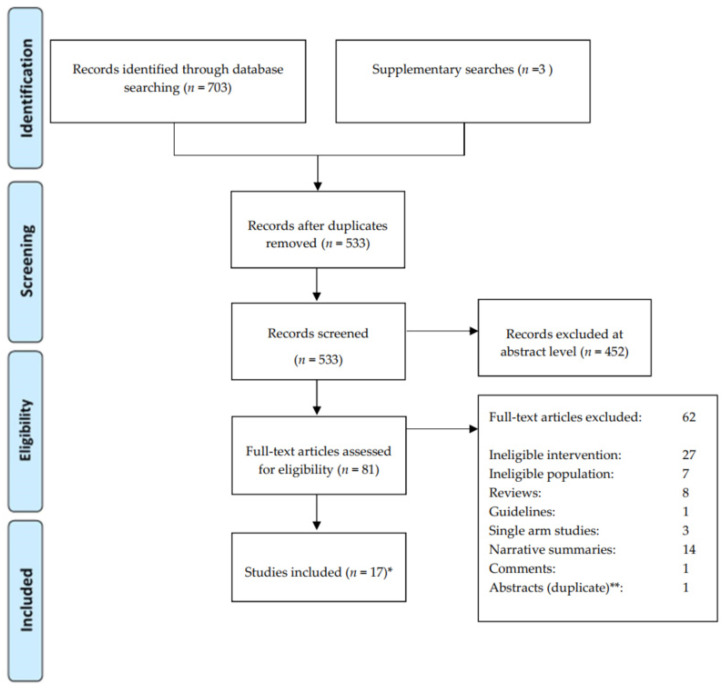
Results of bibliographic literature search and study selection (PRISMA 2009 flow diagram). * 18 studies included, corresponding to 19 published reports. No publication of the ESOSTRATE (Comparison of Systematic Surgery Versus Surveillance and Rescue Surgery in Operable Oesophageal Cancer With a Complete Clinical Response to Radiochemotherapy) protocol could be identified. ** This abstract refers to results reported in the included ESOPRESSO trial (A Randomized Phase III Trial on the Role of Esophagectomy in Complete Responders to Preoperative Chemoradiotherapy for Esophageal Squamous Cell Carcinoma), no additional results are reported.

**Table 1 cancers-13-00429-t001:** Characteristics of randomized and non-randomized controlled trials including protocols.

Study	Study Design (N Centers)	Country/Recruitment Time	Definition of Patients (TNM Staging)	Study Arms	Chemotherapy	Radiotherapy (Total Dose)	Intervention after CRT	Surgery (Time after CRT)	Patients Screened (*n*)	Patients Per Group (*n*)	Age in Years (Range, SD)	Median Follow-Up in Mos (Range)	pCR Rate after CRT (Surgery Group)	On-Demand Surgery Rate during Surveillance
Randomized trials														
Park 2019 [13] (ESOPRESSO)	Phase III RCT (1)	South Korea (2012–2016)	Patients aged 20–75 with histologically confirmed, resectable thoracic squamous cell carcinoma. (cT3-4a, any N, M0 or any T, N+, M0)	I	Capecitabine + Cisplatin	28 × 1.8 Gy (50.4 Gy)	Surveillance	Salvage when indicated	86	18	Median 61 (55–67)	29.9 (13.9–36.3)	69%	33%
				C			Surgery	6–8 weeks		19	Median 62 (56–69)			
Bedenne 2007 [14], Bonnetain 2006 [15] (FFCD 9102)	Phase III RCT (NA)	France (1993–2000)	Patients with resectable epidermoid or adenocarcinoma of the thoracic esophagus and clinical and biologic eligibility for surgery or CRT (T3, N0-1, M0) EAC: 11% ESCC: 89%	I	Cisplatin+5-FU (5 cycles)	NA (45–66 Gy)	Definitive chemoradiation	Salvage by request	451	130	Mean 59.3 (8.9)	24 (NA)	NA	NA
				C	Cisplatin+5-FU (2 cycles)	NA (30–46 Gy)	Surgery	7–9 weeks		129	Mean 57.3 (9.2)			
Stahl 2005 [16]	Phase III RCT (11)	Germany (1994–2002)	Patients up to 70 years with locally advanced squamous cell carcinoma of the upper and mid third of the esophagus. (T3-4, N0-1, M0)	I	5-FU, Leucovorin, Etoposide, Cisplatin	25 × 2 Gy (50 Gy) or 30 × 2 Gy (60 Gy)	Surveillance	Salvage when indicated	189	86	Median 57 (36–71)	60 (NA)	35%	6%
				C		20 × 2 Gy (40 Gy)	Surgery	2–5 weeks		86	Median 57 (37–70)			
Non-randomized trial (treatment based on patient’s decision)														
Fujita 2005 [17]	NRSI (1)	Japan (1994–2002)	Patients with locally advanced squamous cell carcinoma of the thoracic esophagus (defined as a T4 tumor), excluding distant metastasis. All patients were fit for esophagectomy or definitive CRT (T4, N0-1, M0)	I	Cisplatin+5-FU	25 × 2.4 Gy (60 Gy)	Definitive chemoradiation	No surgery	NA	23	NA	51 (NA)	7%	NA
				C			Surgery	4 weeks		30	NA			
Randomized trials (protocols, ongoing) *														
Noordman 2018 [18] (SANO)	Phase III RCT (12)	The Netherlands (2017–2025)	Operable patients with locally advanced, and no clinical evidence of metastatic spread resectable, squamous cell carcinoma or adenocarcinoma of the esophagus or esophagogastric junction. (T1N1 or T2-3N0-1)	I	Carboplatin + Paclitaxel	23 × 1.8 Gy (41.4 Gy)	Surveillance	Salvage when indicated	NA	480	NA	60 planned	NA	NA
				C			Surgery	10–14 weeks			NA			
Jia 2019 [19] (CELAEC)	RCT (3)	China (NA)	Patients aged 18-75 with resectable esophageal squamous cell cancer. (T1bN, M0 or T2-4a, N0-2, M0)	I	Oxaliplatin + Capecitabine, or Cisplatin+5-FU, or Capecitabine alone ^$^	25 × 2 Gy (50 Gy)	Surveillance	Salvage when indicated		196	NA	60 planned	NA	NA
				C		21 × 2 Gy (42 Gy)	Surgery	NA			NA			
Bedenne 2015 (ESOSTRATE) NCT02551458	Phase II/III RCT (NA)	France (NA)	Epidermoid carcinoma or adenocarcinoma of the thoracic esophagus or adenocarcinoma of the esogastric junction (Siewert type I or II) proven histologically (Stage cT2 N1-3 M0 or cT3-T4a N0 or N1-3 M0)	I	Several treatment protocols		Surveillance	Salvage when indicated		57	NA	60 planned	NA	NA
				C			Surgery	NA			NA			

C: control, CELAEC: Chemoradiation versus oesophagectomy for locally advanced oesophageal cancer in Chinese patients, CRT: chemoradiation, EAC: esophageal adenocarcinoma, ESCC: esophageal squamous cell carcinoma, ESOPRESSO: A Randomized Phase III Trial on the Role of Esophagectomy in Complete Responders to Preoperative Chemoradiotherapy for Esophageal Squamous Cell Carcinoma, ESOSTRATE: Comparison of Systematic Surgery Versus Surveillance and Rescue Surgery in Operable Oesophageal Cancer With a Complete Clinical Response to Radiochemotherapy, FFCD: Fédération Francophone de Cancérologie Digestive, I: intervention, N: number, NA: not available pCR: pathological complete response, RCT: randomized controlled trial, SANO: Surgery As Needed approach in Oesophageal cancer patients, TNM: tumor-node-metastases, 5-FU: fluorouracil. * No results published, only trial protocol is available (as of September 2020). ^$^ Patients will receive one of the three possible treatments after a random allocation.

**Table 2 cancers-13-00429-t002:** Characteristics of observational studies.

Study	Study Design (N Centers)	Country/Recruitment Time	Definition of Patient Population (TNM Staging)	Study Arms	Chemotherapy	Radiotherapy (Total Dose)	Intervention	Surgery (Time after CRT)	Patients Observed (*n*)	Age, Years Median (Range)	Follow-Up, Months Median (Range)	pCR Rate after CRT (Surgery Group)	On-Demand Surgery Rate during Surveillance
Gamboa 2020 [20]	Retrospective (cancer registry)	USA (2004–2015)	Patients with nonmetastatic adenosquamous esophageal cancer	Ia	NA	NA	Chemoradiation and Surveillance	NA	74	70/67 *	41 (19–73)	20%	NA
				Ib			Only Surgery		43	67/69 *			
				Ic			Chemoradiation+Surgery		34	59/63 *			
Münch 2019 [21]	Retrospective (1)	Germany (2011–2017)	Patients with histologically proven esophageal squamous cell carcinoma, without distant metastasis. (T1–T4, N+, M0)100% ESCC	I	Carboplatin + Paclitaxel or Cisplatin+5-FU	NA (54 Gy)	Surveillance	Salvage when indicated	55 **	68 (62–74)	25.6 (NA)	38%	NA
				C		NA (41.4 Gy)	Surgery	3.5–12 weeks	40	65 (56–72)			
van der Wilk 2019 [22]	Retrospective (4)	The Netherlands (2013–2016)	Patients with histologically proven, resectable, squamous cell carcinoma or adenocarcinoma of the esophagus or esophagogastric junction without distant metastases, eligible for neoadjuvant chemoradiotherapy (T1–T4, N±, M0) EAC: 72%, ESCC: 27%	I	Carboplatin+ Paclitaxel	23 × 1.8 Gy (41.4 Gy)	Surveillance	Salvage when indicated	31 ***	72.0 (69–77)	27.7 (20–47)	24%	48%
				C			Surgery	12 weeks	67	70.0 (67–73)	34.8 (25–51)		
Castoro 2013 [10]	Retrospective (1)	Italy (1992–2007)	Patients with thoracic esophageal squamous cell carcinoma, without distant metastasis. (NA) 100% ESCC	I	Cisplatin+5-FU	45–50 Gy	Surveillance	Salvage when indicated	38 ^†^	64.7 (57–73)	33.7 (16–81)	69%	37%
				C			Surgery	4–6 weeks	39	58.8 (56–68)	38.8 (19–66)		
Furlong 2013 [23]	Retrospective (1)	Ireland (2000–2007)	Patients with locoregional advanced esophageal adenocarcinoma or squamous cell carcinoma. (NA) EAC: 59%, ESCC: 41%	I	Cisplatin+5-FU	15 × 2.7 Gy (40 Gy)	Surveillance	Salvage when indicated	19 ^§^	Mean 75 (70–83)	NA (2–116)	67%	16%
				C			Surgery	Within 8 weeks	6				
Murphy 2013 [24]	Retrospective (1)	US (2002–2008)	Patients with Stage III resectable adenocarcinoma of esophagus (defined by American Joint Committee on Cancer Staging ), who successfully completed chemoradiation and are eligible for trimodality therapy. (T3N1) 100% EAC	I	Cisplatin+5-FU, or Oxaliplatin+5-FU, or Taxane/Patinum	Median 50.4 Gy	Definitive chemoradiation	No surgery	29 ^$^	70 (48–83)	NA (NA–120)	NA	NA
				C			Surgery	6–8 weeks	114	60 (27–78)			
Taketa 2013 [11]	Retrospective (1)	US (2002–2011)	Patients with resectable gastroesophageal junction or esophageal cancer who are fit for surgery. (T2-T3, N0-N1) EAC: 66%, ESCC: 30%	I	Fluoropyrimidine+ Platinum Compound or Taxane	Median 50.4 Gy	Surveillance	Salvage when indicated	61 **	69 (47–85)	60 (NA)	NA	31%
				C			Surgery	NA	244	59.5 (29–78)			
McKenzie 2011 [25]	Retrospective (cancer registry)	US (1988–2006)	Patients with adenocarcinoma, squamous cell and other types of esophageal cancer, without distant metastasis. (T1-T3, N0–N1, M0) EAC: 38%, ESCC: 55%	I	NA	NA	Unknown if surveillance was available	No surgery	645 ^#^	NA	60 (NA)	NA	NA
				C			Surgery	NA	286	NA			
Wilson 2000 [26]	Retrospective (1)	Canada (1993–1997)	Patients with esophageal adenocarcinoma or squamous cell carcinoma without distant metastases. (T1-T3, N0-N1, M0) EAC: 38%, ESCC: 53%	I	Cisplatin+5-FU	25 × 2 Gy (50 Gy)	Surveillance	Salvage when indicated	24 ^$$^	66 (44–76)	56 (NA)	50%	21%
				C			Surgery	After 12 weeks	6				
Denham 1996 [27]	Retrospective (5)	Australia/ New Zealand (1984-NA)	Patients with lower, middle and upper adenocarcinoma, squamous cell, or other types of esophageal cancer. (NA) EAC: 28%, ESCC: 69%	Ia	Cisplatin+5-FU	30 × 2 Gy (60 Gy)	Definitive chemoradiation	No surgery	169 ^‡^	68.5 (36–91)	60 (NA)	NA	NA
				Ib		15 × 2.3 (35 Gy)	Surgery	9‒12 weeks	92	62 (30–77)			
				Ic		15 × 2 (30 Gy)	Palliative care	No surgery	112 ^‡‡^	69 (36–96)			

C: control, CRT: chemoradiation, EAC: esophageal adenocarcinoma, ESCC: esophageal squamous cell carcinoma, I: intervention, N: number, NA: not available, 5-FU: fluorouracil. * Only separately available for cN0 and cN1 groups (cN0/cN1), ** 26 had cervical cancer, 15 refused surgery and 14 were unfit for surgery. *** All would meet criteria for surgery but refused it. ^†^ 22 refused surgery, 16 were unfit for surgery. ^§^ 8 declined surgery, 11 were unfit for surgery. ^$^ Refused surgery, or surgery was not offered to them. ^$$^ Patients with complete response (negative biopsy). ^#^ Unknown if surgery was an option. ^‡^ No evidence of extra-thoracic disease and unfit for surgery. ^‡‡^ Patients with metastatic disease.

**Table 3 cancers-13-00429-t003:** Characteristics of the survey on patients’ preference.

Study	Study Design (N Centers)	Country/ Recruitment Time	Definition of Patient Population (TNM Staging)	Treatment Choices	Patients in the Survey (*n*)	Age in Years (Range)
Noordman 2018 [28]	Prospective cohort study/survey (2)	The Netherlands (2015–2017)	A survey on patients’ preferences for treatment, with patients who were treated with neoadjuvant chemoradiotherapy according to the CROSS regimen for histologically proven squamous cell or adenocarcinoma of the esophagus or esophagogastric junction. (T2–T4, N0–N3)	Activesurveillance	100	Median 67 (61–72)
				Surgery		

**Table 4 cancers-13-00429-t004:** Outcomes considered and design aspects in the randomized and non-randomized trials including protocols.

Study	Definition of Reported Outcomes	Trial Design	Sample Size Calculation	Result
Park 2019 [13]	DFS (primary outcome): defined as the time between randomization and progression or death from any cause. Progression-free survival: the time between initiation of chemotherapy and progression or death. Time to progression: the time between initiation of chemotherapy and progression. OS: the time between initiation of chemotherapy and death. Failure pattern. Pathologic complete response rate. Treatment outcomes: according to metabolic or clinical response. Quality of life: not available due to a low response rate.	Superiority trial Primary endpoint DFS Aim to show superiority of surgery vs. surveillance with respect to DFS RCT (individual) Randomization of patients with cCR	Assumption: 2-year DFS 70% surgery vs. 50% surveillance, hazard ratio = 0.514 Alpha (two-sided) = 0.05, Power = 80% Required DFS events: 78 Required Patients with cCR: 194 Assumption: cCR rate = 40% Required Patients total: 486 ITT Analysis	Recruitment over 3.5 years Patients total: 86 Patients cCR: 38 (44.2%) Randomized: 37 DFS events:16 Compliance: 68.4% in surgery, 100% in surveillance ITT: 2-year DFS 66.7% surgery, 42.7% surveillance 2-year OS: ca. 70% Reason for early study closure: Low adherence in surgery arm
Bedenne 2007 [14]/Bonnetain 2006 [15]	OS (primary outcome): up to 2 years. Therapeutic Mortality. Length of hospital stay. Recurrence: locoregional, distant, or both, or second cancer at 2 years. Dysphagia and palliative procedures: dysphagia was scored from 1 (asymptomatic) to 5 (no swallowing at all) according to the O’Rourke criteria. Toxicity: graded according to the WHO criteria. Quality of life: evaluated by the Spitzer quality-of-life index, which establishes a score from 0 (worst) to 10 (best) after answering five items in the areas of activity, daily life, health perception, social support, and behaviour.	Equivalence trial Primary endpoint OS Aim to show equivalence of surgery and surveillance with respect to OS RCT (individual) Randomization of patients with cCR/cPR	Equivalence margin 2-year OS difference 10% Alpha (two-sided) = 0.05, Power = 80% Required Patients with cCR/cPR: 360 Assumption: cCR/cPR rate = 75% Required Patients total: 500 ITT and PP Analyses	Recruitment over 7.5 years Patients total: 444 Patients cCR/cPR: ca. 70% Randomized: 259 (58%) OS events: 181 At baseline: 10% cCR, 90% cPR Compliance: 85% in surgery, 97% in surveillance ITT: 2-year OS 33.6% surgery, 38.8% surveillance PP: 2-year OS 37.1% surgery, 36.5% surveillanceReason for early study closure: Interim analysis, due to slow recruitment, data monitoring committee recommendation to stop (surveillance superior)
Stahl 2005 [16]	OS (primary outcome). Progression-free survival. Tumor response: complete remission was defined as no dysphagia, normal barium esophagogram and esophagoscopy, and normal CT scan. Pathologic complete response rate Adverse events: according to National Cancer Institute Common Toxicity Criteria. Postoperative complications.	Equivalence trial Primary endpoint OS Aim to show equivalence of surgery and surveillance with respect to OS RCT (individual) Randomization of patients before start of chemoradiotherapy	Assumption: 2-year OS = 35% Equivalence margin 2-year OS difference 15% (hazard ratio = 0.65) Alpha (one-sided) = 0.05, Power = 80% Required Patients total: 200 After interim analysis of 119 patients: Recalculation: Patients total: 175 ITT Analysis	Recruitment over 8 years Patients total: 189 Randomized: 172 OS events: 132 Compliance: 66% in surgery, 84% in surveillance ITT: 2-year OS 39.9% surgery, 35.4% surveillance
Fujita 2005 [17]	OS: up to 5 years. Response to Chemoradiotherapy/surgery/Pathologic complete response rate. Therapeutic Toxicity: grade 3 or higher according to the National Cancer Institute-Common Toxicity Criteria (1998). Postoperative complications.	NA	NA	NA
Noordman 2018 (Protocol) [18]	OS (primary outcome). The percentage of patients in active surveillance arm who do not undergo surgery. Health-related quality of life: measured with EQ-5D, QLQ-C30, QLC-OG25, and Cancer Worry Scale questionnaires. Clinical irresectability: cT4b rate.Postoperative morbidity/complications for patients who undergo resection: defined by the Esophageal Complications Consensus Group. Postoperative mortality for all patients with clinical complete response who undergo resection: defined as 90 day- and/or in-hospital mortality.Progression-free survival: defined as the interval between randomization and the earliest occurrence of disease progression. Distant dissemination rate. Cost-effectiveness.	Non-inferiority trial Primary endpoint OS Aim to show non-inferiority of surveillance vs. surgery with respect to OS RCT (cluster) Randomization of patients with cCR	Assumption: 3-year OS = 67% Non-inferiority margin 3-year OS difference 15% (hazard ratio = 0.61) Alpha (one-sided) = 0.05, Power = 80% Required Patients with cCR: 300 Assumption: cCR rate = 50% Required Patients total: 600 ITT and PP Analyses	Correspondence with BPL Wijnhoven on status in 12/2019: Patients total: 461 Patients randomized: 160 Compliance: 75% in surgery, 100% in surveillance Trial ongoing
Jia 2019 (Protocol) [19]	OS (primary outcome): at 2 and 5-years of follow up. DFS: at 2 and 5-years of follow up. Treatment-related adverse events. Quality of life: using the Quality of Life Questionnaire-Core 30 (QLQ-C30 version 3.0, in Chinese) and the supplemental Quality of Life Esophageal Module 18 Questionnaire (QLQ-ES18, in Chinese) developed by the European Organization for Research and Treatment of Cancer (EORTC).	Superiority trial Primary endpoint OS Aim to show superiority of surveillance vs. surgery with respect to OSRCT (individual) Randomization of patients before start of chemoradiotherapy	Assumption: 5-year OS 29.4% surgery vs. 50% surveillance Alpha (two-sided) = 0.05, Power = 80% Required Patients total: 192	Trial ongoing
Bedenne 2015 (Protocol) [14]	Proportion of surviving patients (Time Frame: 1 year after randomization) DFS (Time Frame: Up to 5 years)	Superiority trial Primary endpoint DFS Aim to show superiority of surgery vs. surveillance with respect to DFS RCT (individual) Randomization of patients with cCR	Assumption: 2-year DFS 45% surgery vs. 30% surveillance, hazard ratio = 0.66 Alpha (two-sided) = 0.05, Power = 85% Required DFS events: 224 Required Patients with cCR: 260 Assumption: cCR rate = 40% Required Patients total: 600 ITT and PP Analyses	Trial ongoing

OS: overall survival, DFS: disease-free survival, cCR: clinical complete response, cPR: clinical partial response, ITT: intention to treat, PP: per protocol.

**Table 5 cancers-13-00429-t005:** Outcomes considered in the observational studies/attributes that were considered in the survey.

**Study**	**Definition of Reported Outcomes**
Gamboa 2020 [20]	OS (primary outcome). Pathologic complete response rate.
Münch 2019 [21]	OS (primary outcome). Treatment failure. Progression-free survival.
van der Wilk 2019 [22]	OS: time between date of diagnosis and date of all-cause death or last follow-up. Proportion of radical resection: defined as no tumor cells at the proximal, distal, and circumferential resection margin. 30- and 90-day postoperative mortality. Frequency and severity of postoperative complications: severity of post-operative complications was defined according to the Clavien–Dindo classification. Type of complication was classified according to the definitions of the Esophagectomy Complications Consensus Group. Rate and timing of distant dissemination: defined as the time between date of diagnosis and date of detection of distant metastases. Progression-free survival: defined as the time between date of diagnosis and date of detection of progression or last follow-up (with censoring afterward). Progression was defined as the development of distant metastases or development of irresectable locoregional recurrence.
Castoro 2013 [10]	OS. Disease Recurrence rate. Post-operative complications: anastomotic leak rate, post-actinic esophageal stenosis rate.
Furlong 2013 [23]	OS. Disease Recurrence rate. Hospital Mortality: hospital mortality following surgery.
Murphy 2013 [24]	OS: calculated from the day of first treatment until the last known date of follow-up or date of death. DFS: calculated from the day of first treatment until the first known date of disease recurrence or last known date of follow-up or date of death. Clinical response: evaluated using the following criteria: (1) no evidence of disease progression on post- neoadjuvant chemoradiation regional or distant PET; (2) negative post- neoadjuvant chemoradiation biopsy; and (3) decrease in local pre- and post-neoadjuvant chemoradiation standard uptake value (SUV) on PET-CT ≥ 35%.
Taketa 2013 [11]	OS. 3-year relapse-free survival.
McKenzie 2011 [25]	OS: calculated from the date of diagnosis to the date of death or the date of last follow-up.
Wilson 2000 [26]	Median disease specific survival. Post-treatment complete histologic response rate. Grade 3 and 4 toxicity frequencies: as defined by the Common Toxicity Grades of the National Cancer Institute of Canada.
Denham 1996 [27]	Cause specific survival. Complete tumor clearance. Incidence of relapse.
Noordman 2018 [28]	Relevant attributes that influenced the choice of esophagectomy versus active surveillance after neoadjuvant chemoradiation among surveyed patients: 5-year OS: the risk that a regrowth is detected at an unresectable stage during active surveillance (T4b or distant metastases). Health-related quality of life: presented on a visual analogue scale ranging from 0 to 100, where 100 represents the best health status. It was measured using the five-level version of the EuroQol Five Dimensions questionnaire (EQ-5D-5L™; EuroQol Group, Rotterdam, the Netherlands). The risk that esophagectomy is still necessary: the risk that residual disease is missed during the initial response evaluation, but is detected at a resectable stage during active surveillance. The frequency of clinical examinations: using endoscopy and PET–CT.

OS: overall survival, DFS: disease-free survival.

## Data Availability

Data is contained within the article or Appendix A. The primary data presented in this systematic review are available in the primary studies cited in the reference list.

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
