# Peer review of "Post-Neoadjuvant Surveillance and Surgery as Needed Compared with Post-Neoadjuvant Surgery on Principle in Multimodal Treatment for Esophageal Cancer: A Scoping Review"

_cancers, 2021, doi:10.3390/cancers13030429_

Round 1
Reviewer 1 Report
A through evaluation of existing papers concering the subject. A clear problem is inclusion of studies more than 15 years old and including only a few studies with junctional cancers.
In the western world the esophageal cancers (ESC) are declining and the important cancers are the EAC, still increasing in number and located at the EG junction. Thus:"Moreover, high-level evidence from RCTs is almost entirely missing for atients with EAC, therefore there is an urgent need to perform further RCTs regarding surveillance and surgery as needed with modern diagnostic and multimodal treatment strategies." I completely agree.
I miss informations on: "An expert medical sciences librarian developed
the search strategies, which were then peer-reviewed." should be added
Reviewer 2 Report
#1. page 2 “treated with neoadjuvant chemotherapy (nCT) or neoadjuvant chemoradiation (nCRT) followed by surgery on principle [1].” & ref-1 “2013;24 Suppl 6:vi51”? >> but in the latest version [Ann Oncol . 2016 Sep;27(suppl 5):v50-v57, https://www.esmo.org/guidelines/gastrointestinal-cancers/oesophageal-cancer, accessed Dec 22, 2020], neoadjuvant chemotherapy was used only for adenocarcinoma but not squamous cell carcinoma [see Fig 1 in Ann Oncol . 2016 Sep;27(suppl 5):v50-v57]
#2. page3 “(prospective registration identifier of the clinical trial: DRKS 00022801)” >> The reviewer had searched https://www.drks.de/drks_web/navigate.do?navigationId=search&reset=true but failed to find DRKS 00022801
#3. page 3“six RCTs, one NRS, ten observational studies and one survey” vs “*17 studies included, with a total of 18 published reports” in Fig -1 >> (6+1) in table-1 plus 10 (table-2) plus 1 (table-3) = 18 > 17 in Fig 1 [17 studies with 18 published reports], so please clarify the number of included studies
#4. page 12: 1st paragraph in discussion “post-neoadjuvant surveillance and surgery as needed in complete clinical responders” & page 14 conclusion “post neoadjuvant surveillance and surgery as needed is feasible for complete clinical responders.” >> should “complete clinical responders” be kept or deleted? In the reviewer’s mind, this [keeping “complete clinical responders”] would be the most reasonable research question [to limit the study population to only the complete clinical responders]. However, this was not compatible with the title and eligibility criteria [see “population” in section 5.2] of the current manuscript, because “complete clinical responders” was not required in some of the included studies [such as FFCD 9102…., see section 2.1.3 “Randomization was carried out after the response evaluation in the ESOPRESSO- and FFCD 9102-trials in case of cCR (and clinical partial response in the FFCD 9102-trial) either to immediate surgery or surveillance with surgery as needed”]
#5. page 12: 2nd paragraph in discussion “Not one study using nCT protocols was identified” >> but nCT was allowed in the ESOSTRATE-trial ? [see section 2.3.2 “The ESOSTRATE-trial includes patients with cCR after neoadjuvant chemoradiation or neoadjuvant chemotherapy without specifying a treatment protocol”]
#6. page 12: 2nd paragraph in discussion “The direct comparison of nCT and nCRT is also missing until the results of the ongoing ESOPEC-trial (NCT02509286) will be available [35]” >> how about “Dis Esophagus . 2019 Feb 1;32(2)= NCT01362127”? “BMC Cancer . 2017 Jun 28;17(1):450= NCT03001596”? “Jpn J Clin Oncol . 2013 Jul;43(7):752-5= JCOG1109”?
#7. page 14 “This scoping review is based on a prospectively published protocol (“currently under review in the peer-reviewed journal BMJ Open”)” >> So how can the reviewer[s] +/- readers know what are the differences between the manuscript under review in BMJ Open vs the current manuscript under review in Cancers?
#8. page 14 “Appendix” >> The reviewer can’t find the details of Appendix in the downloaded PDF for review
#9. page 15 “Risk of bias (RoB) assessment is not part of a scoping review, hence, RoB was not assessed [39].” >> Ref-19 was written in Germany. It would be great if the authors could prove a reference written in English.
Reviewer 3 Report
A scoping review on post neoadjuvant surveillance and surgery in esophageal cancer, related to an upcoming RCT. The topic is one of the most highly debated topics in the field of esophageal cancer and is of very high interest in the researcher community. The review is generally well-written and organized with some non-idiomatic or uncommon English expressions (for example "pathohistological" is should be written as histopathological, and "inhomogenous" should be heterogenous). Adequate references to previous work have been made. Congratulations for your work and good luck with your trial, hope it will shed more light on the subject!
Specific comments:
-English proofreading is recommended for the manuscript as a whole
-The methods are short, so I would recommend moving methods between introduction and results (the IMRAD organization) if not specifically required in this order by the journal
-Tables are somewhat cumbersome (especially 1 and 2) to read due to the information overload and narrow columns. This might be more of a journal typesetting problem, but some divisions to more tables or using some selected abbreviations might help.
Round 2
Reviewer 2 Report
# section 5.1 “This scoping review is based on a prospectively published protocol in “BMJ Open” (reference will be added as soon as published [in Feb 2021])” >> Suggest to re-submit the revision after this BMJ Open paper was accessible, or enclose that paper as a supplementary material